# Attention-Enhanced Lightweight Architecture with Hybrid Loss for Colposcopic Image Segmentation

**DOI:** 10.3390/cancers17050781

**Published:** 2025-02-25

**Authors:** Priyadarshini Chatterjee, Shadab Siddiqui, Razia Sulthana Abdul Kareem, Srikant Rao

**Affiliations:** 1Department of Computer Science and Engineering, Koneru Lakshmaiah Education Foundation, Hyderabad 500075, Telangana, India; cseshadabsiddiqui@gmail.com; 2Old Royal Naval College, University of Greenwich, Park Row, London SE10 9LS, UK; razia.sulthana@greenwich.ac.uk; 3MNJ Institue of Oncology and Regional Cancer Center, Hyderabad 500004, Telangana, India; srikanthsapthagiri@yahoo.com

**Keywords:** cervical cancer, image segmentation, contextual information, loss function, multi-scale feature extraction, refinement module

## Abstract

Cervical cancer screening through computer-aided diagnosis is often challenging due to inaccurate segmentation and incomplete boundary detection in colposcopic images. This study presents a lightweight segmentation model that integrates dual encoder backbones (ResNet50 and MobileNetV2) to efficiently extract features. The model uses a novel attention module to improve segmentation accuracy and an atrous spatial pyramid pooling (ASPP) module for multi-scale contextual learning. The experimental validation demonstrates high performance, with 97.56% training accuracy, 96.04% validation accuracy, and a Dice coefficient of 98.71%, outperforming the existing approaches. This research contributes to advancing precise and efficient colposcopic image segmentation, improving diagnostic accuracy in clinical applications.

## 1. Introduction

As the fourth most frequent cancer among women globally, cervical cancer continues to be a major global health concern. In 2022, there were about 660,000 new cases and 350,000 deaths from the disease [1]. The three main screening techniques recommended by the World Health Organization (WHO) are liquid-based cytology (LBC), traditional Papanicolaou (Pap) smear, and high-risk human papillomavirus (HPV) DNA testing. However, the latter two are often complex and costly to administer. In many developing countries, colposcopy-directed biopsy is commonly utilized for diagnosis, with individuals showing positive cytology or HPV test results being referred for colposcopy as per the American Society for Colposcopy and Cervical Pathology (ASCCP) guidelines [2]. Accurate colposcopic diagnosis requires practitioners to precisely identify acetowhite epithelium characteristics, a skill heavily reliant on clinical experience. In regions with limited medical resources, the scarcity of experienced clinicians and substantial screening workloads present significant challenges [3]. Computer-aided diagnosis (CAD) has advanced significantly as a result of advances in artificial intelligence (AI), improving diagnostic consistency and accuracy while lightening the strain of medical personnel. Notably, AI applications in colposcopy have primarily focused on broad lesion classification and detection of high-grade squamous intraepithelial lesions (HSIL) or more severe conditions, with relatively few studies addressing precise lesion segmentation. Accurate segmentation is crucial, providing essential guidance for colposcopists in lesion classification and biopsy site selection, thereby playing a vital role in cervical cancer diagnosis [4,5]. This study proposes a deep-learning-based segmentation model utilizing a dual encoder architecture that combines ResNet50 and MobileNetV2 backbones. Skip connections from both encoders preserve contextual information across network layers, integrating high-resolution spatial details from MobileNetV2 with deep semantic features from ResNet50. This dual-stream approach enhances the model’s ability to address intricate edges, improve segmentation precision, and effectively detect regions of interest in colposcopic images [6,7,8]. To address class imbalance in medical image segmentation, the model employs a hybrid loss function that combines Tversky loss and Dice loss. This combination optimizes both overall lesion detection and precise boundary delineation, proving effective for segmenting irregular lesions in imbalanced datasets [9,10,11,12]. Adding an atrous spatial pyramid pooling (ASPP) module, which uses parallel atrous convolutions with different dilation rates to capture multi-scale contextual information, improves feature extraction even more. This approach is particularly beneficial for accurately segmenting irregular and faintly edged lesions in imbalanced datasets [13,14]. Additionally, a lightweight refinement module is integrated to enhance low-level feature representation by focusing on spatial details during decoding. This module applies convolutional layers followed by batch normalization to refine feature maps, emphasizing boundary details and subtle texture variations. Its design ensures minimal computational overhead while improving segmentation accuracy, particularly for complex structures or fine-grained details [15,16]. The proposed architecture aims to improve feature extraction and integrate upsampling concatenation for colposcopic image segmentation. The model demonstrates excellent generalization capabilities, performing effectively on both private and public datasets. The specific contribution of this study is as follows:We integrated a custom loss function that combines Dice loss and Tversky loss. This combined approach enhances segmentation precision by penalizing false negatives and false positives differently, as defined by the Tversky loss, while maintaining global accuracy with Dice loss. The combination ensures robustness in handling imbalanced datasets and improves segmentation performance on challenging colposcopic images.A lightweight refinement module was incorporated to enhance feature map quality by leveraging convolutional layers and depthwise separable convolutions. Batch normalization and channel-wise attention mechanisms were added to refine spatial and semantic features. This module helps resolve blurred boundaries and incomplete segmentation regions in colposcopic images.A lightweight ASPP block was introduced to capture multi-scale contextual information by applying depthwise dilated convolutions at varying rates. The ASPP block also includes a global pooling layer to integrate broader spatial context. This design enables the model to detect both fine details and larger anatomical structures in colposcopic images, significantly improving segmentation across varying scales.The model incorporates a dual encoder architecture, combining feature maps from ResNet50 and MobileNetV2 backbones. Skip connections ensure spatial and contextual information from each encoder is preserved and integrated seamlessly. This dual-encoder approach enriches the model’s capability to detect complex textures and edges in colposcopic images, addressing challenges of texture variability and fine feature reconstruction.An attention mechanism was applied in the decoder to refine features propagated from skip connections. By employing attention modules, the model highlights significant regions while suppressing irrelevant information, enhancing segmentation precision. The decoder structure ensures effective reconstruction of fine-grained features, further improving the segmentation of complex regions in colposcopic images.

## 2. Materials and Methods

### 2.1. Materials

The International Agency for Research on Cancer provided a collection of 918 pictures of CIN1, CIN2, and CIN3 lesions [17]. These images were captured using binocular colposcopes with integrated cameras and supplemented with metadata containing 920 entries in a .csv file. The participants had a mean age of 27 years, with 2.55 average sexual partners and a median age of first intercourse at 17. Moreover, 15% were smokers, 65% used hormonal contraceptives, and 70% had HPV infections. Regarding diagnoses, 97% had CIN, while 2.1% had cancer.

The dataset includes images taken using Lugol’s iodine, acetic acid, and normal saline. For consistency, the images were organized in a standard sequence: two images post-acetic acid application, one with a green filter and one with iodine solution, as illustrated in Figure 1. If the sequence was disrupted, filenames were adjusted accordingly.

The dataset was classified into CIN1, CIN2, and CIN3 and further categorized based on transformation zone visibility. The first category includes images where both inner and outer boundaries of the transformation zone are visible. The second category consists of images where the transformation zone is partially visible, both inside and outside the cervix. The third category contains images where the transformation zone is entirely within the cervix and not visible. A 1:1:1 ratio was maintained across these categories.

The LabelMe annotation tool [18] was used for manual annotation, with annotations validated by a radiation oncologist at MNJ Institute of Oncology, Hyderabad. The annotated polygonal regions were used to create binary masks stored in .json format and organized by CIN grade for further analysis, as depicted in Figure 2.

This structured dataset serves as the primary dataset for this study, ensuring consistency and reliability in colposcopic image segmentation.

#### Primary Dataset and Preprocessing of the Primary Dataset

A total of 860 images (278 CIN1, 286 CIN2, 296 CIN3) were used in this study. To standardize input dimensions, the images and their corresponding binary masks were resized to 256 × 256 pixels, with pixel intensity values normalized to [0,1]. Binary masks corresponding to each image were resized to match the input image dimensions, ensuring proper alignment as shown in Figure 3. Backbone-specific preprocessing for ResNet50 and MobileNetV2 included mean subtraction and standard deviation scaling. Data augmentation was performed using the Albumentations library [19] to enhance diversity and prevent overfitting. Transformations—random rotations, flips, brightness/contrast adjustments, and elastic deformations—ensured consistent alignment between images and masks. The dataset was expanded to 2000 images while preserving the original class proportions (646 CIN1, 665 CIN2, 689 CIN3), as shown in Figure 4. With a small class imbalance (2%), the combined loss function (Dice + Tversky) ensures accurate segmentation. Dice loss optimizes region overlap, while Tversky loss balances false positives and false negatives, maintaining precision without biasing toward dominant classes. This augmentation strategy balances data sufficiency and computational efficiency, ensuring robust model training while preventing excessive redundancy. Augmented images and masks were organized into CIN1, CIN2, and CIN3 folders for streamlined training and evaluation.

### 2.2. Masking Architecture

The proposed masking algorithm generates binary masks from JSON annotation files by iterating over the annotated shapes and creating filled polygons on a blank mask. Each annotation is processed as follows:The JSON file is parsed to extract polygon coordinates defined for each shape.A blank grayscale image (mask) is created with dimensions matching the corresponding input image.The polygons are drawn on the mask using the annotation points, ensuring the coordinates lie within the mask’s bounds.The mask is saved as a separate image for subsequent use in segmentation tasks.

This algorithm ensures that masks are generated precisely as per the annotations while retaining flexibility for different image dimensions and formats. Additionally, the system handles cases where annotations might be partially out of bounds, ensuring robustness. The masks are validated by an experienced radiation oncologist who is our fourth author.

#### Advantages over Mask R-CNN

While Mask R-CNN is a powerful instance segmentation framework, the proposed algorithm has the following advantages for specific use cases like this:Efficiency: The algorithm directly utilizes existing JSON annotations to create masks, avoiding the need for computationally expensive model training and inference.Annotation Consistency: Since the masks are created directly from ground truth annotations, they preserve the precise shapes and boundaries of annotated regions, which is critical for tasks requiring high-fidelity segmentation.Customizability: The algorithm allows easy handling of complex shapes or irregular annotations, making it adaptable for diverse datasets.Resource Requirements: Unlike Mask R-CNN, which requires significant computational resources for training and inference, this method can be executed efficiently on standard hardware, making it suitable for resource-constrained environments.

This approach is particularly advantageous for scenarios where annotations are already available, and the goal is to generate precise binary masks without the overhead of training a deep learning model. Figure 5 shows the process of masking that we employed for generating the binary mask.

All the masks generated are validated by an experienced radiation oncologist, Dr. R. Srikanth, who is an eminent professor in the department of oncology of MNJ Institute of Oncology—Regional Center, Hyderabad.

### 2.3. Model Description

Our model, inspired by [3,20], employs ResNet50 and MobileNetV2 backbones to balance high-level feature extraction with computational efficiency. To enhance segmentation performance, the model uses a combined loss function integrating Dice and Tversky losses. Dice loss and Tversky losses ensure pixel-wise accuracy and optimize the overlap and address class imbalance. Additionally, a lightweight atrous spatial pyramid pooling (ASPP) block captures multi-scale contextual information using depthwise separable convolutions with varying dilation rates and global average pooling. This efficient design reduces computational overhead while effectively handling objects of different sizes. Grad-CAM++ visualizations provide interpretability by highlighting areas that influenced the model’s decisions, fostering clinician trust. We have posted our model in GitHub (We utilized GitHub Desktop (Version 2022-11-28), developed by Tom Preston-Werner, Chris Wanstrath, P. J. Hyett, and Scott Chacon, headquartered in San Francisco, CA, USA) [21].

#### 2.3.1. Deep Learning Model as the Backbone

##### ResNet50

The vanishing gradient issue that frequently occurs during the training of deep neural networks was explicitly addressed by ResNet50, a deep residual network that was first shown in [22]. ResNet introduces the concept of residual learning, where the model learns residual mappings instead of directly learning deference mappings, making it suitable for very deep architectures. ResNet50 consists of 50 layers, structured as a series of convolutional layers with skip connections. These skip connections enable effective gradient flow through the network, preserving high-level feature representation at deeper layers. The residual blocks consist of identity mappings that bypass the intermediate layers, allowing the network to optimize better. One study [23] adopted ResNet50 for the classification of brain tumors. The main difference between the standard ResNet50 architecture and the modified ResNet50 used in this model is the exclusion of the fully connected layers and the selection of specific intermediate feature maps to suit the segmentation task. While the baseline ResNet50 is designed for classification tasks with a fully connected top layer producing class probabilities, the modified ResNet50 is repurposed as a feature extractor by setting include-top = False. This ensures that the output comprises spatially rich feature maps instead of classification logits. Additionally, specific layers (conv2-block3-out, conv3-block4-out, conv4-block6-out, and conv5-block3-out) are extracted, representing feature maps at different levels of the network. These layers capture low-level to high-level features, enabling multi-scale feature representation that is essential for segmentation tasks. This modification leverages ResNet50’s hierarchical architecture while ensuring compatibility with the encoder–decoder structure of the segmentation model. By selectively using intermediate layers, the computational efficiency of the network is maintained without requiring further structural changes.

Reason for Selection: Compared to other backbone architectures like VGGNet or DenseNet, ResNet50 is more efficient due to its skip connections, which reduce the computational complexity and improve training convergence. It provides high-level feature representation that is crucial for tasks like image segmentation. VGGNet lacks skip connections and is computationally heavier, while DenseNet introduces excessive inter-layer connections, leading to increased memory requirements.

Mathematical Representation: The residual block in ResNet50 can be formulated as:(1)$$y=G(x,\left\{W_{i}\right\})+x\;$$
where
x represents the input to the residual block,$G(x,\left\{W_{i}\right\})$ denotes the residual function defined by the weights $\left\{W_{i}\right\}$,y represents the output of the block.

The operation $x+G(x,\left\{W_{i}\right\})$ in Equation (Equation 1) ensures that the network focuses on learning residual functions instead of direct mappings, which aids in maintaining efficient gradient propagation.

##### MobileNetV2

A lightweight convolutional neural network called MobileNetV2 [24] was created for effective deployment on smartphones with limited resources. Compared to conventional convolutions, it drastically lowers the computational cost by using depthwise separable convolutions. MobileNetV2 introduces inverted residuals and linear bottlenecks to preserve feature diversity in low-dimensional embeddings. The architecture consists of a sequence of bottleneck residual blocks. Each block comprises depthwise convolution for spatial filtering, pointwise convolution for feature combination, and linear bottlenecks to prevent non-linearity in compressed representations. One of the recent study uses MobileNetV2 for breast tumor detection [25]. The main difference between the standard MobileNetV2 architecture and the modified MobileNetV2 used in the proposed model lies in its adaptation for feature extraction in a segmentation task. While the baseline MobileNetV2 is designed for classification tasks with a fully connected top layer, the modified MobileNetV2 excludes the classification layers (include-top = False) and outputs spatial feature maps instead of class probabilities. This adjustment makes it suitable for integration into the encoder–decoder framework of the segmentation model. In the proposed model, specific intermediate layers (block-1-expand-relu, block-3-expand-relu, block-6-expand-relu, and block-13-expand-relu) are extracted to capture multi-scale features at varying depths of the network. These layers provide a progression from fine-grained, low-level details to coarse, high-level semantic representations. This hierarchical feature extraction is crucial for handling the diverse scales and complexities inherent in segmentation tasks.

Reason for Selection: Compared to architectures like Inception or EfficientNet, MobileNetV2 is computationally more efficient and has lower memory requirements. Its inverted residuals allow for high-quality feature extraction even with limited computational resources, making it an ideal candidate for combining with ResNet50 in this hybrid segmentation model.

Mathematical Representation: The depthwise separable convolution in MobileNetV2 can be represented as Equation (Equation 2):(2)$$y=\left(X*W_{\mathrm{depthwise}}\right)*W_{\mathrm{pointwise}}\;$$
where
X is the input feature map,$W_{\mathrm{depthwise}}$ and $W_{\mathrm{pointwise}}$ are the depthwise and pointwise convolution filters, respectively,∗ denotes the convolution operation.

The inverted residual block is formulated as(3)$$y=x+F(x,\left\{W_{i}\right\})\;\mathrm{if}\;\mathrm{residual}\;\mathrm{connection}\;\mathrm{exists},$$
where $F(x,\left\{W_{i}\right\})$, as in Equation (Equation 3), includes depthwise separable convolutions and bottleneck transformations.

##### Why Combine ResNet50 and MobileNetV2?

ResNet50 provides rich, high-level feature representations due to its deep architecture and skip connections, making it effective for capturing global contextual information.MobileNetV2 contributes lightweight, efficient computations, ensuring the model remains computationally feasible while capturing low-level spatial details.The combination leverages the strengths of both architectures, creating a robust model suitable for high-quality segmentation with reduced computational overhead.

#### 2.3.2. Loss Functions

In the proposed model, a combined loss function is employed, integrating Dice loss and Tversky loss. Each loss function addresses different aspects of the segmentation task to ensure robust optimization.

##### Dice Loss

Dice loss, Equation (Equation 4), is used in colposcopy image segmentation because it directly optimizes the Dice coefficient, which measures the overlap between predicted and ground truth regions. This is particularly important in colposcopy, where the boundaries of abnormalities, such as lesions, need to be precisely segmented. By focusing on maximizing region overlap, Dice loss enhances the model’s ability to accurately delineate abnormal tissue, which is critical for effective diagnosis and intervention [26,27]:(4)$$\mathrm{Dice}\;\mathrm{Loss}=1-\frac{2\sum y\hat{y}+\epsilon }{\sum y+\sum \hat{y}+\epsilon }$$
where ϵ is a small constant added to avoid division by zero.

##### Tversky Loss

Tversky Loss [27] is used in colposcopy image segmentation to address the inherent class imbalance between abnormal regions (e.g., lesions) and the surrounding healthy tissue. By introducing weighting factors α and β, as in Equation (Equation 5), Tversky loss allows the model to prioritize minimizing false negatives (missed abnormalities) while managing false positives, ensuring accurate segmentation of small or subtle abnormal regions. This adaptability is crucial for reliable identification of abnormalities in colposcopy images:(5)$$\mathrm{Tversky}\;\mathrm{Loss}=1-\frac{\sum y\hat{y}+\epsilon }{\sum y\hat{y}+\alpha \sum y\left(1-\hat{y}\right)+\beta \sum \left(1-y\right)\hat{y}+\epsilon }$$

Here, α and β control the penalties for false negatives and false positives, making it particularly useful for imbalanced datasets.

##### Combined Loss

The combined loss function combines BCE, Dice, and Tversky losses, Equation (Equation 6), to balance pixel-level accuracy, region-level segmentation quality, and robustness against class imbalance [27]:(6)CombinedLoss=DiceLoss+TverskyLoss

#### 2.3.3. Attention Mechanism

The attention mechanism in the proposed model incorporates a squeeze-and-excitation (SE) block to enhance feature representation for colposcopy image segmentation. This mechanism reweights channel features by emphasizing informative features related to abnormalities, such as lesions, while suppressing irrelevant ones. The SE block is mathematically represented as in Equation (Equation 7):(7)$$\mathrm{SE}\;\mathrm{Output}=\sigma (W_{2}\;\mathrm{ReLU}\left(W_{1}\;\mathrm{GAP}\left(X\right)\right))\cdot X$$
where GAP denotes global average pooling, $W_{1},W_{2}$ are learnable weights, and σ is the sigmoid function. This attention mechanism ensures the model focuses on critical features, improving segmentation accuracy for detecting subtle abnormalities in colposcopy images.

#### 2.3.4. Decoder

The decoder in the proposed model fuses features extracted from ResNet50 and MobileNetV2 backbones through concatenation at multiple scales. This multi-scale fusion enhances the model’s ability to capture both fine-grained and high-level contextual information. The decoder uses transpose convolutions to upsample feature maps and restore spatial resolution as in Equation (Equation 8):(8)y=X⊗W
where ⊗ denotes transposed convolution. Upsampling operations are employed to ensure the output matches the input image dimensions, enabling precise pixel-level segmentation of abnormalities in colposcopy images. This design allows the model to effectively delineate lesions while maintaining computational efficiency.

#### 2.3.5. Lightweight Atrous Spatial Pyramid Pooling (ASPP)

The lightweight ASPP module in the proposed model captures multi-scale contextual information, which is crucial for colposcopy image segmentation. By using dilated convolutions with varying dilation rates, ASPP can effectively detect abnormalities of different sizes, such as small lesions or larger abnormal regions, within the cervix.

The ASPP output is computed as in Equation (Equation 9):(9)$$\mathrm{ASPP}\;\mathrm{Output}=\mathrm{Concat}({\mathrm{Conv}}_{\mathrm{rate}=1},{\mathrm{Conv}}_{\mathrm{rate}=6},{\mathrm{Conv}}_{\mathrm{rate}=12},\mathrm{Global}\;\mathrm{Avg}.\;\mathrm{Pool})$$

This design combines spatially detailed features with global contextual information, ensuring that the model captures fine-grained structures while retaining a global perspective. The lightweight nature of the ASPP module maintains computational efficiency, making it well-suited for precise and resource-efficient colposcopy image segmentation.

#### 2.3.6. Grad-CAM++ Visualization

Grad-CAM++ is utilized in the proposed architecture to provide interpretability for colposcopy image segmentation [28]. This visualization technique highlights the regions of the input image that contribute most to the model’s predictions, enabling clinicians to understand and validate the segmentation results.

The implementation computes a weighted combination of feature maps from a specified layer (e.g., conv5_block3_out) based on the gradients of the predicted output with respect to these feature maps. The process involves
Computing gradients of the predicted class score with respect to the feature maps using tf.GradientTape.Weighting the feature maps by the averaged guided gradients.Aggregating the weighted feature maps to generate a class activation map (CAM).Normalizing the CAM to create a heatmap that overlays onto the original image, highlighting critical regions.

This technique is particularly beneficial for colposcopy image segmentation, as it provides visual explanations of the model’s focus, aiding in the identification of abnormal regions, such as lesions. By ensuring transparency and interpretability, Grad-CAM++ enhances the trustworthiness of the proposed model in clinical applications.

### 2.4. Proposed Model Architecture

The main contribution of this study is the development of a deep convolutional neural network (CNN) optimized for segmenting colposcopy images. The model integrates advanced components like attention mechanisms, dual backbone architectures, and multi-scale feature aggregation to address the challenges of colposcopy image segmentation. Every aspect of the model has been thoughtfully designed to ensure precise and efficient segmentation:Dual Backbone Architecture: The model employs ResNet50 and MobileNetV2 as backbone networks. ResNet50 extracts high-level semantic features with its deep architecture and residual connections, while MobileNetV2 efficiently captures lightweight features. Feature maps from both backbones are concatenated at multiple scales to combine their strengths.Attention Mechanism: Squeeze-and-excitation (SE) blocks are used to create channel-wise attention, reweighting feature maps to focus on critical visual patterns in colposcopy images, such as vascular abnormalities and epithelial changes.Lightweight ASPP: The lightweight atrous spatial pyramid pooling (ASPP) module uses global average pooling and dilated convolutions with different dilation rates to capture multi-scale contextual information. This guarantees that anomalies of different sizes are successfully recorded.Decoder Design: The decoder reconstructs the spatial resolution of the segmented masks using transpose convolutions and upsampling layers. It fuses features from ResNet50 and MobileNetV2 through concatenation at different scales, ensuring precise delineation of lesions.Loss Function: A combined loss function integrates Dice and Tversky losses to ensure pixel-wise accuracy, region-level overlap optimization, and robustness against class imbalance, making the model effective for segmenting small or subtle abnormalities.Input Preprocessing: Colposcopy images are resized to 256×256 pixels, and data augmentation techniques such as rotations and flips are applied in real-time to improve generalization and reduce overfitting.Training Strategy: With a batch size of 16 and a learning rate of 1 × 10^−4^, the model is trained across 300 epochs. Early halting and learning rate scheduling are used to maximize efficiency and avoid overfitting.Evaluation Metrics: The model is evaluated using metrics such as the Dice coefficient, intersection over union (IoU), and Hausdorff distance, ensuring comprehensive assessment of segmentation quality.Grad-CAM++ for Explainability: The areas of the input image that most influence the model’s predictions are visualized using grad-weighted class activation mapping (Grad-CAM++). This enhances interpretability and allows clinicians to validate the model’s focus on medically relevant areas.

Each component of the model contributes to its robust and interpretable performance, ensuring its applicability to real-world colposcopy image segmentation tasks.

The pseudocode for these stages may be found in Algorithm 1. The architecture of the feature extraction, two backbones, the atrous spatial pyramid pooling block (ASPP), the squeeze-and-excitation block (SE), and the decoder are depicted in Figure 6. Table 1 provides a summary of the tensor transformations. Theorem 1 describes the theorem that may be inferred from the aforementioned stages. The proposed model’s structure for colposcopic image segmentation, emphasizing feature refinement and attention, retains a U-shaped profile but incorporates a nested structure for more precise feature merging across different scales, as illustrated in Figure 7.
**Algorithm 1** U-Net with ResNet50 and MobileNetV2 for Colposcopy Segmentation **function** FORWARD_SEGMENTATION(X,Y)    **Step 1: Data Loading and Preprocessing**    $X_{\mathrm{resized}}\leftarrow \mathrm{Resize}\;\mathrm{images}\;\mathrm{to}\;256\times 256\times 3$    $Y_{\mathrm{resized}}\leftarrow \mathrm{Resize}\;\mathrm{masks}\;\mathrm{to}\;256\times 256$    Divide data into training, validation, and test sets.    **Step 2: Feature Extraction with Dual Backbones**    Use **ResNet50** to extract multi-scale features:    $F_{\mathrm{ResNet}}$ = [conv2_block3, conv3_block4, conv4_block6, conv5_block3]    Use **MobileNetV2** to extract lightweight features:    $F_{\mathrm{MobileNet}}$ = [block1_expand, block3_expand, block6_expand, block13_expand]    **Step 3: Feature Fusion and Attention Mechanism**    Resize MobileNetV2 features to match ResNet50.    Combine feature maps: $F_{\mathrm{combined}}\leftarrow \mathrm{Concatenate}\left(F_{\mathrm{ResNet}},F_{\mathrm{MobileNet}}\right)$    Apply Squeeze-and-Excitation (SE) blocks for attention.    **Step 4: Lightweight ASPP for Multi-scale Context**    $F_{\mathrm{ASPP}}\leftarrow \mathrm{Apply}\;\mathrm{dilated}\;\mathrm{convolutions}\;\mathrm{with}\;\mathrm{rates}\;1,\;6,\;\mathrm{and}\;12\;\mathrm{and}\;\mathrm{global}\;\mathrm{pooling}$    **Step 5: Decoder for Spatial Reconstruction**    **for** each decoder block (4 stages) **do**     $F_{\mathrm{upsampled}}\leftarrow \mathrm{Transpose}\;\mathrm{convolution}\;\mathrm{and}\;\mathrm{upsampling}$     Concatenate corresponding encoder features for multi-scale fusion.    **end for**    Final segmentation output: $Y_{\mathrm{pred}}\leftarrow \mathrm{Conv}2D(1)\;(\mathrm{Sigmoid}\;\mathrm{activation})$    **Step 6: Training and Validation**    Define **Combined Loss**: Dice + Tversky    Train model using Adam optimizer and early stopping.    Evaluate on validation/test sets using metrics: Dice coefficient, IoU, precision, recall.    **Step 7: Inference Performance on Different Hardware**    Measure inference time per image:    **Colab Pro (TPU V2-8)**: $t_{\mathrm{TPU}}=5.2$ milliseconds    **Standard CPU (Intel i5-1035G1)**: $t_{\mathrm{CPU}}=28.4$ milliseconds    **Step 8: Grad-CAM++ for Explainability and Cohen’s Kappa Calculation**    Generate Grad-CAM++ heatmaps to highlight ROI influence.    Compute Cohen’s Kappa (κ) for agreement with expert-annotated ground truth:
$$\kappa =\frac{p_{o}-p_{e}}{1-p_{e}}$$    where $p_{o}$ is the observed agreement, and $p_{e}$ is the expected agreement by chance.    **return** $Y_{\mathrm{pred}},\mathrm{Grad}\text{-}\mathrm{CAM}\;\mathrm{heatmaps},\kappa$ **end function**


**Theorem** **1.**
*Let M be a deep convolutional neural network with dual backbone encoders (ResNet50 and MobileNetV2), an attention module, a lightweight atrous spatial pyramid pooling (ASPP) block, and a decoder for segmentation tasks. The model M is trained using a combined loss function L, defined as*

$$L\left(y,\hat{y}\right)=L_{\mathrm{Dice}}\left(y,\hat{y}\right)+L_{\mathrm{Tversky}}\left(y,\hat{y}\right),$$

*where $L_{\mathrm{Dice}}$ and $L_{\mathrm{Tversky}}$ are Dice loss and Tversky loss, respectively. This architecture and loss function combination ensures pixel-wise accuracy, overlap optimization, and robustness to class imbalance, achieving optimal segmentation performance.*


**Proof.** The model M combines the strengths of high-level and low-level feature extraction, feature refinement, and efficient decoding, while the combined loss L optimizes segmentation accuracy.1. Model Architecture M:
*Encoders:* ResNet50 and MobileNetV2 extract hierarchical high-level and low-level features, respectively. These features are resized and fused to form a rich feature representation.*Attention Module:* Enhances critical channel features through a squeeze-and-excitation mechanism:
$$\mathrm{Attention}\;\mathrm{Output}=\sigma (W_{2}\;\mathrm{ReLU}\left(W_{1}\;\mathrm{GAP}\left(X\right)\right))\cdot X,$$
where $W_{1},W_{2}$ are learnable weights, *σ* is the sigmoid activation, and GAP denotes global average pooling.*Lightweight ASPP:* Captures multi-scale context using depthwise separable convolutions with varying dilation rates:
$$\mathrm{ASPP}\;\mathrm{Output}=\mathrm{Concat}({\mathrm{Conv}}_{\mathrm{rate}=1},{\mathrm{Conv}}_{\mathrm{rate}=6},{\mathrm{Conv}}_{\mathrm{rate}=12},\mathrm{GAP}).$$*Decoder:* Combines high-level features from ASPP and attention-enhanced features from the encoder to progressively restore spatial resolution using transpose convolutions and concatenation.2. Combined Loss Function L: The loss function L optimizes the segmentation output $\hat{y}$ by balancing pixel-wise accuracy, region-level overlap, and class imbalance:
*Binary Crossentropy Loss:**Dice Loss:*$$L_{\mathrm{Dice}}\left(y,\hat{y}\right)=1-\frac{2\sum_{i=1}^{N}y_{i}{\hat{y}}_{i}+\epsilon }{\sum_{i=1}^{N}y_{i}+\sum_{i=1}^{N}{\hat{y}}_{i}+\epsilon },$$
where ϵ>0 prevents division by zero, maximizing overlap between predicted and ground truth masks.*Tversky Loss:*$$L_{\mathrm{Tversky}}\left(y,\hat{y}\right)=1-\frac{\sum_{i=1}^{N}y_{i}{\hat{y}}_{i}+\epsilon }{\sum_{i=1}^{N}y_{i}{\hat{y}}_{i}+\alpha \sum_{i=1}^{N}y_{i}\left(1-{\hat{y}}_{i}\right)+\beta \sum_{i=1}^{N}\left(1-y_{i}\right){\hat{y}}_{i}+\epsilon },$$
where α,β≥0 balance false negatives and false positives, addressing class imbalance.3. Training Optimization: By minimizing L, the network M learns to optimize pixel-wise prediction, region-level segmentation accuracy, and class-imbalance robustness. The decoder progressively refines feature maps to output a segmentation mask $\hat{y}$ that matches the ground truth *y*.Thus, the architecture M combined with the loss function L ensures optimal segmentation performance in terms of accuracy and robustness. □

### 2.5. Time Complexity of the Proposed Model

#### 2.5.1. Comparative Study of Time Complexity: Proposed Model vs. Models of [3,20]

##### Time Complexity of Models of [3,20]

The models depicted in [3,20] include components such as
First Model: CLS-Net with EfficientNet-B3 and ASPP
–*Backbone:* EfficientNet-B3 has a computational complexity of $O(d_{\mathrm{EfficientNet}})$, where $d_{\mathrm{EfficientNet}}$ depends on the depth and width multipliers of the network.–*ASPP:* The atrous spatial pyramid pooling module involves dilated convolutions, contributing an additional complexity of O(k), where *k* represents the dilation rates and feature map size.–*Upsampling and Mapping:* Decoder complexity includes upsampling and fusion operations (O(u)).Second Model: RUC-U^2^-Net
–*Recursive Blocks:* R^2^SU blocks in the encoder and decoder stages involve recursive convolutions, contributing $O(r\cdot d_{R^{2}\mathrm{SU}})$, where *r* is the number of recursions and $d_{R^{2}\mathrm{SU}}$ represents the depth of each block.–*Oblique Attention:* The oblique attention connection (OACM) adds complexity proportional to the feature map resolution (O(a)).

##### Time Complexity of the Proposed Model

The proposed model includes
*Backbones:* ResNet50 and MobileNetV2 have a combined complexity of $O(d_{\mathrm{ResNet}}+d_{\mathrm{MobileNet}})$, depending on the depth and feature map sizes of the respective layers.*Attention Mechanism:* The SE block introduces O(c), where *c* depends on the number of channels.*ASPP Block:* Similar to the CLS-Net, the ASPP block contributes O(k).*Decoder:* The decoder uses transposed convolutions and concatenations with complexity O(u).

##### Comparison and Justification

CLS-Net vs. Proposed Model: EfficientNet-B3 in CLS-Net is more computationally efficient than the combined ResNet50 and MobileNetV2 backbones in the proposed model. However, both use similar ASPP and decoder structures, making the overall complexity of the proposed model slightly higher.RUC-U^2^-Net vs. Proposed Model: RUC-U^2^-Net employs recursive convolutions and OACM, significantly increasing its complexity compared to the proposed model. Recursive operations in RUC-U^2^-Net scale non-linearly with feature map size, making it more computationally intensive for large images.

##### Conclusions

Among the three models, CLS-Net is computationally the most efficient due to its lightweight EfficientNet-B3 backbone. RUC-U^2^-Net has the highest complexity due to its recursive blocks and attention connections. The proposed model balances complexity and segmentation accuracy with moderately higher computational demands than CLS-Net but lower than RUC-U^2^-Net.

### 2.6. Ablation Study

By using only the ResNet50 backbone for feature extraction and keeping the attention mechanism and lightweight atrous spatial pyramid pooling block (ASPP) module for improving feature refining and multi-scale context aggregation, this ablation model uses a simplified architecture. The model’s computational complexity and memory overhead are decreased by eliminating the MobileNetV2 backbone, which essentially cuts down on training and inference time. This lightweight configuration ensures faster training while maintaining high segmentation performance, making it suitable for real-time applications in practical scenarios. This structure is depicted in Figure 8. The details of the proposed structure without MobileNetV2 with a stride of 2 is shown in Figure 9.

## 3. Results

### 3.1. Experimental Environment and Parameter Settings

The model discussed herein was implemented using the TensorFlow 2.15.0 deep learning framework, with Keras 2.15.0 and Python 3.10.12. Experiments were conducted in a Colab Pro environment equipped with TPU V2-8 hardware for accelerated training. The host system ran Windows 10 Home Single Language OS, as detailed in Table 2. The training configuration included 300 epochs, an initial learning rate of 0.0001, and a batch size of 16. The dataset, augmented to 2000 images, was split into an 8:1:1 train–validation–test ratio.

For inference, the model demonstrated efficiency across different hardware setups. On **Colab Pro (TPU V2-8)**, it achieved an inference time of **5.2 milliseconds per image**, while on a **Standard CPU (Intel i5-1035G1)**, it processed images in **28.4 milliseconds**, as depicted in Algorithm 1. Given that real-time clinical diagnostics require inference times below **100 milliseconds**, the model meets practical applicability standards for clinical settings.

### 3.2. Metrics

The study uses several evaluation metrics to analyze the model’s segmentation accuracy, as shown in Table 3. The reasons for selecting these metrics are described under the head of “Description” in the table.

### 3.3. Experimental Results

#### 3.3.1. Quantitative Analysis

In this section, the proposed models, the hybrid net and the ablation, are quantitatively compared with six other mainstream medical image segmentation models using the dataset introduced in this study. The results on the validation sets are presented in Table 4. The models cited as [20,29,30,31,32,33] are existing models from Section 4.7 and are also identified as research gaps. The models developed in this research outperform the comparison group across all six evaluation metrics. It is important to mention here how the models cited as [20,29,30] are modified so that they could be executed on our primary dataset in Section 2.1 and in the same experimental setup as mentioned in Section 3.1.

Ref. [29]: To transition the unsupervised learning model for colposcopy image segmentation into a supervised learning model, the following modifications are made. The adjustments ensure the model leverages labeled data (ground truth masks) to improve segmentation performance.

##### Key Adjustments for Supervised Learning

Ground Truth Labels: Replaced the pseudo-masks used in the unsupervised setting with ground truth segmentation mask.Loss Function: Replaced the reconstruction loss (used in the unsupervised setting) with supervised segmentation-specific loss functions as in our model.Training Process: Simultaneously train both the encoder and decoder of the U-Net using labeled images and their ground truth segmentation masks. Loss backpropagation is guided by the direct comparison of the segmentation output with the ground truth masks.Metrics for Supervised Evaluation:
Evaluate the segmentation performance using standard metrics:
–Pixel Accuracy (PA): Measures the ratio of correctly predicted pixels.–Mean Intersection over Union (MIoU): Reflects the model’s ability to overlap with ground truth regions.–Dice Coefficient: Highlights overlap between predicted and actual masks.–Hausdorff Distance: Assesses the maximum boundary discrepancy between predicted and ground truth masks.Learning Rate Strategy:
Use a uniform learning rate for both encoder and decoder since there is no need to prioritize the encoder in supervised settings.

##### Implementation in TensorFlow

The model implementation remains consistent with the U-Net architecture, modified for supervised learning. Below are the implementation steps:Define the Supervised U-Net Model:
–Structure the model to include the encoder, decoder, and skip connections.–Use TensorFlow and Keras to define the layers.Prepare the Dataset:
–Organize the dataset with input images and their corresponding ground truth masks.–Split the dataset into training, validation, and test sets.Use the Adam optimizer with an initial learning rate of 0.0001.Train the model using the fit() function with the training and validation splits.

Ref. [30]: This model is primarily used for comparing the lesion segmentation of colposcopy images before and after acetic acid solution. The model used is the baseline U-Net so there was no need of any modification. We have used our primary dataset in Section 2.1 as is and analyzed the metrics as mentioned in Table 4, apart from the qualitative evaluation.

Ref. [20]: This model also was executed as it is, except for the faster RCNN, we have used our primary dataset in Section 2.1 with the corresponding masks as an input to the model. The model was analyzed on the metrics as in Table 4, apart from the qualitative evaluation.

#### 3.3.2. Qualitative Analysis

##### Prediction of the Ground Truth by the Models

Figure 10 illustrates the segmentation performance analysis of colposcopic images using the two proposed models in this study on the test set, bench marked against six other existing models on the validation dataset. We have used overlay, which is a visualization technique used to superimpose one image or data layer onto another, enabling a clear comparison or highlighting specific information. In the context of medical imaging, overlays are often employed to combine the original image (e.g., colposcopic images) with segmentation masks or annotations. This allows for a better understanding of how well a model has segmented or classified regions of interest by visually aligning predictions with the original image.

#### 3.3.3. Feature Visualization

We visualized the features extracted by our proposed method through heatmaps, where the intensity of the red color indicates a higher contribution of a region to the model’s final classification, while the blue color represents a lower contribution. In essence, the model relies more heavily on the red-highlighted areas for its decisions. This is demonstrated in Figure 11 on the test samples. Specifically, Figure 11a–d depict the post-acetic acid image, the ground truth segmentation, the predicted segmentation, and the Gradcam++ visualization, respectively, captured on the validation dataset. The original image is taken in acetic acid, but the color of the image changed due to normalization and the application of GradCam++.

#### 3.3.4. Quantitative Analysis of Feature Visualization

Cohen’s Kappa (κ) was used to evaluate the agreement between GradCam++ heatmaps and expert-annotated regions. As shown in Table 5, the proposed model achieved the highest κ of 0.94, indicating near-perfect agreement with clinical experts. This surpasses traditional models such as U-Net (κ=0.85) and the CLS Model (κ=0.88), demonstrating the model’s superior lesion localization and segmentation accuracy.

#### 3.3.5. Ablation Experiment

As indicated in Table 6 on the validation set, we performed ablation experiments on our primary dataset Section 2.1 and in the same experimental environment Section 3.1 to assess the contributions of various components in order to assess the suggested model. We started with the baseline ResNet50+decoder model and saw small gains when MobileNetV2 encoder features and attention modules were added. The integration of the lightweight ASPP module further enhanced the model’s ability to capture multi-scale features, resulting in significant performance gains. Finally, the proposed model, which combines all these components with a carefully designed loss function, achieved the highest segmentation accuracy across all metrics, including PA, MPA, MIoU, and FWIoU. These results demonstrate the effectiveness of each module in addressing segmentation challenges in colposcopic images.

#### 3.3.6. Performance of the Model Without Data Augmentation on the Primary Dataset

##### Quantitative Analysis

As indicated in Table 7 on the validation set, we performed ablation experiments on our primary dataset as in Section 2.1 without augmentation and in the same experimental environment Section 3.1 to assess and the compare the performance of the model. Section 2.1 shows that the data imbalance in the original primary dataset is 2.09% and in the primary augmented dataset is 2.00%, both being very minor; there is not much difference in the metrics obtained outlined in Table 7.

##### Qualitative Analysis

Figure 12 illustrates the model’s predicted masks using both augmented and non-augmented datasets. Given the minimal class imbalance in both cases, the model’s ROI predictions remain largely consistent, with no significant variations in segmentation performance.

#### 3.3.7. Loss Function Experiments

For this study, the dataset was meticulously curated. A subset of 250 images with clear and well-defined boundaries within the ROI was filtered to form the refined sample, as depicted in Figure 13a. In contrast, another 300 images with blurred and intricate boundaries within the ROI, categorized as challenging samples, are shown in Figure 13b. Table 8 and Table 9 summarize the results of applying the Tversky loss function, Dice loss function, and the combination of Tversky loss and Dice loss to the proposed segmentation model. Table 8 highlights the outcomes for simple samples, whereas Table 9 emphasizes the results for challenging samples. Both of these tables are on the validation samples. These findings are further illustrated in Figure 14 for a visual comparison. 

#### 3.3.8. Generalization Experiments

The LGG Segmentation Dataset for brain MRI segmentation [34] and the secondary dataset, the Malhari dataset for cervical cancer segmentation [35], are publicly available datasets used to assess the generalizability of the suggested models. Comparative experiments were performed against other main stream medical segmentation models, with the results presented below. Both of these datasets are licensed under public domain.

##### LGG Segmentation Dataset

Brain MR images and manual FLAIR abnormality segmentation masks are included in this dataset. They match 110 individuals with at least the fluid-attenuated inversion recovery (FLAIR) sequencing and genomic cluster data available who were part of the lower-grade glioma collection in The Cancer Genome Atlas (TCGA). Eleven thousand pictures and masks are included. The distribution of images and masks being 7417 and 3531, respectively. There were no pre-processing and augmentation required. The dataset was used as it is. The dataset was randomly divided into a test set and a training set in a 2:8 ratio. We have used the same experimental environment as in Section 3.1. Quantitative comparisons with other existing medical segmentation models [20,29,30] underscore the superior performance of the proposed model across all six evaluation metrics, as shown in Table 10 on the validation set. Figure 15 illustrates the segmentation results for brain MRI images, comparing the two proposed model with three others on the test set. The results highlight the CLS Model [20] and our proposed model’s remarkable precision in segmentation, closely aligning with the manually annotated regions of interest (ROI (column b)). Notably, models c, d, and g in the third, fourth, and in the seventh column (ablation study) of Figure 15 exhibit significant information loss during segmentation. In contrast, the CLS Model [20] and the proposed model (column e and column f) leverage an adaptive attention mechanism, enhancing the allocation of weights between vessels and background during the learning process, thereby effectively mitigating the information loss observed in other models. This experiment validates the proposed model’s and the CLS Model’s [20] robust generalization capability, underscoring their practical applicability.

##### Malhari Dataset

We have acquired the secondary dataset from Kaggle, named Malhari [35], containing 2790 images. This dataset also had images captured in three different solutions: Lugol’s iodine, acetic acid, and normal saline. In the original dataset, CIN1, CIN2, and CIN3 were distributed as follows: 900, 930, and 960, respectively. We have followed the preprocessing step described in Section 2.1, except the image augmentation. The images were annotated using the LabelMe annotation tool [18] to define polygonal regions of interest, which were categorized into folders based on CIN grades (CIN1, CIN2, and CIN3). The images were used as is without elimination of any particular image in the dataset. We have used the same experimental environment as in Section 3.1. The dataset was randomly split into training and test sets in a 8:2 ratio. Comparative analyses with other existing medical segmentation models [20,29,30] highlight the superior performance of the proposed model across all six evaluation metrics, as detailed in Table 11 on the validation data. Figure 16 presents segmentation results for the Malhari cervical cancer data set, comparing the two proposed model against three other models on the test data. The results emphasize the exceptional accuracy of both the CLS Model [20] and the proposed model in delineating regions of interest (ROI), as seen in column b of Figure 16. In contrast, models c, d, and g (third, fourth, and seventh columns (representing an ablation study), respectively) demonstrate considerable information loss during segmentation. Both the CLS Model [20] and the proposed model (columns e and f) employ an adaptive attention mechanism that optimally balances weight allocation between the target regions and background, effectively addressing the limitations observed in other models. These findings confirm the robust generalization capabilities of the CLS Model [20] and the proposed model, reinforcing their practical applicability in medical image segmentation tasks.

## 4. Literature Review

The existing literature on the methods used to segment different medical images and non medical images will be thoroughly reviewed and critically analyzed in this part. By examining the previous studies in this field, we want to identify information gaps and limits and highlight how our study could help to solve these problems, offering important new perspectives on this urgent public health issue of early detection of cervical cancer. The existing literature has been explored from databases such as IEEE Xplore, MDPI, PubMed, SpringerLink, ScienceDirect, ACM, and other relevant sources.

### 4.1. TraditionalMethods

Xu, Chen, and their research team compared the OTSU method with a rapid multithreshold image segmentation approach based on histogram analysis. Their findings demonstrated that their method outperformed the OTSU method in terms of computational efficiency and segmentation quality [36]. Similarly, Hu, Zhao, and their team proposed an algorithm for multithreshold image segmentation inspired by Fick’s law. Despite being a widely used and specialized technique, its practical application is largely restricted to medical imaging due to its high computational requirements and challenges in achieving satisfactory segmentation outcomes [37]. On a different front, Qamar, Malyshev, and collaborators developed a hybrid method combining a convolutional neural network (CNN) and a random forest (RF) classifier to analyze and classify bacterial spore layers from transmission electron microscopy (TEM) images. Their model achieved 73% accuracy, 64% precision, 46% sensitivity, and 47% F1-score on the test data [38].

### 4.2. U-Net and Its Variants

Zannah, Bashar, and their team have carried out comparative research on u-net and its variant for the segmentation of dental X-ray images and achieved that the vannila u-net received the highest result of performance, achieving an accuracy of 95.56% and an IoU score of 88% on the validation set [39]. Kande. Ravi, and their research team have presented the multi-scale residual (MSR) U-Net model for fundus image segmentation. In comparison to the existing methods, the experimental findings consistently show better or equivalent performance, obtaining higher accuracy, F1 score, and area under the receiver operating characteristic (AUC). These results demonstrate enhanced efficacy in separating blood arteries with different thicknesses, even in difficult situations with conflicting lesions, complex vessel shapes, and varied contextual information [40]. Chen, Kim, and their collaborators introduce a multi-convolutional channel residual spatial attention U-Net (MCRSAU-Net) specially designed for industrial and medical image segmentation. This net achieved remarkable performance on three datasets with average dice coefficients of 0.7755, 0.7651, and 0.8958, respectively, on the three datasets [41]. Lu, Tian, and their associates introduced HMSAM-UNet. HMSAM integrates the hierarchical attention mechanism and the inception module through residual connections. The hierarchical attention mechanism highlights important regions by learning attention weights, significantly improving the model’s capacity to identify critical areas for more precise localization and segmentation of target structures in CT images [42]. Zhou, Siddiquee, and their partners have introduced UNet++ and evaluated its performance against u-net and its variations in a number of medical image segmentation tasks, including polyp segmentation in colonoscopy videos, liver segmentation in abdominal CT scans, nuclei segmentation in microscopy images, and nodule segmentation in low-dose CT scans of the chest. According to their experiment, UNet++ with deep supervision outperforms U-Net and wide U-Net by an average IoU gain of 3.9 and 3.4 points, respectively [43].

### 4.3. Deep Learning Architectures with Feature Refinement

Peng, Chen, and their team have proposed dual attention modules based on the Deeplabv3+ network in the DCN-Deeplabv3+ architecture, a road segmentation technique that improves segmentation accuracy while lowering model parameters and processing requirements [44]. He, Jiang, and their associates suggested a lossless image compression-based end-to-end BAA model called the squeeze-and-excitation deep residual network (SE-ResNet). Their approach fared better than the baseline models, according to tests conducted on a publicly available dataset [45]. Rashid, Aslam, and their team proposed a modified squeeze-and-excitation ResNet (SE-ResNet) architecture to categorize pigment markings in color fundus images of retinitis pigmentosa (RP). This variation improves feature representation by combining the strong attention mechanism of the SE block with the effectiveness of residual skip connections. To find the ideal layer configuration that strikes a balance between computational economy and performance measurements, the SE-ResNet model was refined. The RIPS dataset, which contains pictures of people with RP diagnoses at different stages, was used to train the suggested model [46].

### 4.4. Attention-Mechanism-Based Models

Armand, Bhattacherjee, and their team used four separate data sets to assess the efficacy of several designs (UNet, UNetR, TransUNet, and Swin-UNet), with and without transformers, in medical imaging [23]. Tsai, Chang, and their associates propose a new UU-Mamba model that combines an uncertainty-aware loss function and the sharpness-aware minimization (SAM) optimizer with the U-Mamba model. By locating flat minima in the loss landscape, SAM reduces overfitting and improves generalization. To improve segmentation accuracy and resilience, the uncertainty-aware loss function integrates pixel-, region-, and distribution-based loss approaches. The approach outperforms cutting-edge models like TransUNet, Swin-Unet, nnUNet, and nnFormer when tested on the ACDC cardiac data set. The efficacy of the method in cardiac MRI segmentation is demonstrated by its high Dice similarity coefficient (DSC) and low mean squared error (MSE) scores [47]. Veni, Gupta, and their team proposed an innovative hybrid deep learning model for acne classification that integrates the attention mechanism of the convolutional block attention module (CBAM) for enhanced feature selection, combined with the VGG16 and CapsNet architectures. This model autonomously identifies critical features using CBAM within the VGG16 framework and subsequently uses CapsNet for feature classification. The model delivered exceptional performance metrics across all datasets, achieving 100% precision, 99% F1 score, 100% recall, 99% accuracy, 100% specificity, and a kappa score of 97.87% [48]. Fu, Chen, and their associates proposed an augmented U-network that combines a modified convolutional block attention module (CBAM) and an improved pyramid pooling module (PPM) within a U-Net architecture. Although the updated CBAM incorporates attention methods during upsampling, the enhanced PPM improves feature extraction during the downsampling stage. To improve segmentation performance, the network also makes use of innovative RGB training. According to experimental results, segmentation accuracy (Dice, IoU, MAE, BFscore) and training efficiency have improved significantly, surpassing both conventional U-shaped networks and sophisticated models such as U-Net++, U2-Net, ResU-Net, ResU-Net++, and UNeXt [49].

### 4.5. Models with Advanced Loss Functions

In order to segment ischemic stroke lesions in fluid-attenuated inversion recovery (FLAIR) and diffusion weighted imaging (DWI) images, Sinha, Bhatt, and their team introduced a unique deep learning model called EnigmaNet. EnigmaNet’s encoder and decoder architectures include a dual attention mechanism and novel Genesis-k blocks. To improve performance, a modified weighted focal-Tversky–Dice (wFTD) loss is used. The model demonstrates a Dice score of 0.8965, sensitivity of 0.8776, and specificity of 0.9866 for FLAIR test pictures and a Dice score of 0.8423, sensitivity of 0.8452, and specificity of 0.9754 for DWI images when tested in the ISLES-2015 public dataset [50]. Predanan, Suzuki, and their associates presented a two-stage pipeline for urinary stone segmentation. In the first step, a U-Net model creates a map that locates urinary organs in complete abdominal X-ray images. In order to correct class imbalance and enlarge the training dataset, this map is then used in the second stage to construct partitioned input images and for stone embedding enhancement. In order to segment urinary stones on partitioned inputs, the U-Net model is trained using a combination of photos containing genuine stones and images embedded in synthetic stones. The lesion size imbalance is further addressed using an inverse weighting technique to the focal Tversky loss function [51].

### 4.6. Colposcopy-Specific Models

Inspired by the DenseNet model, Saini, Bansal, and their partners designed ColpoNet due to its computational efficiency compared to other architectures. To evaluate the applicability of the method, it was tested and verified on the National Cancer Institute dataset and compared to deep learning models including AlexNet, VGG16, ResNet50, LeNet, and GoogleNet. According to experimental data, ColpoNet outperformed other state-of-the-art approaches in terms of accuracy, achieving 81.353% [52]. Prakash, Ayyasamy, and their team introduced an updated version of the DenseNet-121 deep-learning model designed for identifying pediatric pneumonia in medical scans. Enhancements include the integration of batch normalization, max pooling, and dropout layers to improve model accuracy. Batch normalization scales and normalizes the activations of preceding layers to achieve a mean of zero and variance of one, reducing internal variability during training, accelerating the training process, improving model stability, and improving generalization. Max pooling reduces the number of parameters, increasing computational efficiency, while dropout prevents overfitting by reducing neuron codependence, allowing the network to learn more robust and adaptive features [53].

In conclusion, the advances in medical image segmentation discussed in this review underscore the transformative potential of deep learning in this domain. Traditional segmentation methods, while foundational, often struggle with accuracy and adaptability in complex scenarios. The emergence of U-Net and its numerous variants has marked a significant step forward, offering robust solutions for various medical imaging tasks. Further enhancements, such as deep learning architectures with feature refinement and attention-mechanism-based models, have demonstrated the ability to capture intricate details and improve segmentation precision. In addition, models that incorporate advanced loss functions have addressed challenges such as class imbalance and variability of the size of the injury, pushing the limits of segmentation precision. Colposcopy-specific models, tailored to unique imaging requirements, exemplify the power of specialization in achieving superior outcomes. These advancements not only pave the way for optimizing neural networks for greater diagnostic precision but also open new avenues for integrating machine learning innovations to enhance patient care and clinical decision making.

### 4.7. Gaps Identified in Other Existing Models and Baseline Models and Corrective Measures Taken

Extensive research has revealed several gaps that the proposed method seeks to address. The approaches were carefully selected for implementation on the primary dataset. Table 12 summarizes these identified gaps, the limitations of the existing methods, and the corrective measures introduced by the proposed approach.

## 5. Discussion

### 5.1. Cervical Colposcopy Analysis and Model Development

Cervical colposcopy analysis traditionally involves three key stages: **identifying the cervical region, extracting lesion-specific features, and conducting diagnostic assessment**. This study focuses on **automating the region of interest (ROI) detection**, reducing dependency on manual intervention and enhancing the efficiency of colposcopic image analysis.

The proposed model addresses **common segmentation challenges**, such as **edge lesions, obstructions, artifacts, and blurriness**, improving diagnostic efficiency. The integration of **ResNet50 and MobileNetV2** for feature extraction, **squeeze-and-excitation (SE) blocks** for global features, and **lightweight atrous spatial pyramid pooling (ASPP)** for contextual information enhances generalization. Additionally, a **hybrid loss function (Dice + Tversky)** ensures adaptability to **both simple and complex images**, as demonstrated in **Algorithm 1, Table 1, and Figure 7 and Figure 8**.

### 5.2. Performance Evaluation and Model Refinement

Initial models pretrained on **ImageNet** showed **overfitting and poor generalization**, with validation accuracies around **93% (Table 4)**. To improve robustness, **attention mechanisms (SE blocks, ASPP)** were incorporated, leading to superior performance over models lacking these enhancements (**Table 12, Figure 10, Figure 15 and Figure 16**). Models without attention mechanisms (**[29,30]**) exhibited **loss of intricate lesion features**, whereas models with **SE and ASPP** demonstrated better ROI localization (**columns f and g in Figure 10**).

To further validate **segmentation accuracy**, we used **six evaluation metrics** (**Table 3**) across **existing segmentation models** and our proposed framework. The results (**Table 4**) confirmed the advantage of **SE and ASPP blocks**, which extracted **global and contextual features**, reducing overfitting and enhancing interpretability.

### 5.3. Baseline Model Comparisons

Baseline models (**[31,32,33]**) lacked attention mechanisms, resulting in **average performance** on **both segmentation metrics and qualitative visualization (Table 4, Figure 10, Figure 15 and Figure 16)**.

An **ablation study** (**Table 6, Figure 8 and Figure 9**) using **only ResNet50** demonstrated its **limitation in extracting high-level features**. The absence of **MobileNetV2** resulted in **suboptimal generalization**, leading to **higher information loss and degraded performance on different datasets** (**Table 10 and Table 11, Figure 15 and Figure 16**). The **addition of MobileNetV2** significantly improved segmentation accuracy by capturing fine-grained lesion details (**Table 6**).

### 5.4. Impact of Loss Functions

We evaluated the effects of **Dice loss, Tversky loss, and their combination** on **simple and complex images (Table 8 and Table 9, Figure 14)**.
**Dice loss** improved **pixel accuracy (PA) and mean pixel accuracy (MPA)**, making it effective for **clear lesion boundaries**.**Tversky loss** enhanced **intersection over union (IoU) and FW-IoU**, performing well on **unclear boundaries**.**The combined loss (Dice + Tversky)** balanced both effects, ensuring **robust generalization across diverse colposcopic datasets**.

### 5.5. Generalization Across Different Datasets

To assess **model robustness**, we tested it on **two independent datasets** (**Section 3.3.8**), comparing against **[20,29,30]**. Our **proposed model and [20]** demonstrated superior performance (**Table 10 and Table 11, Figure 15 and Figure 16**) due to **enhanced attention mechanisms**. Models without attention suffered from **significant information loss**, leading to **poor generalization**.

## 6. Conclusions

This study presents a robust deep-learning model for colposcopic image segmentation, integrating ResNet50 and MobileNetV2 backbones with attention mechanisms (SE blocks, ASPP) and a hybrid loss function (Dice + Tversky). The model effectively addresses challenges such as incomplete edge segmentation and loss of fine-grained details, outperforming existing segmentation methods across multiple evaluation metrics.

Experimental validation confirms significant performance improvements: the model achieves a Dice coefficient of 98.71%, IoU of 97.55%, and a Hausdorff distance of 1346.75 pixels, exceeding prior methods (Table 4). Additionally, Grad-CAM++ heatmaps show that the model effectively localizes regions of interest, with a Kappa coefficient of 0.94 indicating near-perfect agreement with expert annotations (Table 4).

The model demonstrates computational efficiency for real-time clinical applications, achieving an inference time of 5.2 ms per image on TPU and 28.4 ms on a standard CPU, well below the 100 ms threshold required for real-time diagnostics (Algorithm 1, Table 1). These findings support its feasibility for integration into automated cervical cancer screening workflows.

Despite these advancements, certain limitations remain. The model’s computational complexity necessitates high-performance hardware for deployment, and its generalizability across diverse imaging modalities requires further validation. Future research will focus on:Developing lightweight architectures optimized for resource-constrained environments.Exploring self-supervised learning to reduce reliance on annotated data.Integrating multi-modal imaging to enhance segmentation robustness.Investigating federated learning approaches for privacy-preserving medical collaborations.

**Clinical Implications:** As we are collaborating with the MNJ Institute of Oncology Regional Center, Hyderabad, we have demonstrated the model’s practical applicability in clinical workflows. The model has been tested using Pap Smear stains converted to digital images and has achieved a segmentation accuracy of 92.01%, 89.00% specificity, 91.08% sensitivity, 93.62% Dice coefficient, and 91.26% IoU.

In general, this study presents a significant step forward in AI-driven cervical cancer screening, providing a clinically relevant and computationally efficient solution for colposcopic image segmentation.

## Figures and Tables

**Figure 1 cancers-17-00781-f001:**
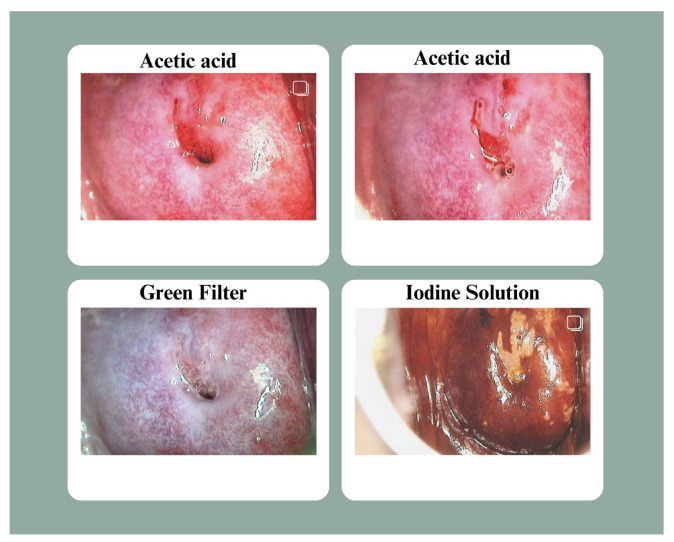
Dataset Sample: As per the above image, the acetic acid solution is used in the first two photos, the iodine solution is applied in the last column, and the image from the second-last column is examined via a green lens. This sample is of the same patient and belongs to Class 1 (cervical neoplasia of the intraepithelial layer).

**Figure 2 cancers-17-00781-f002:**
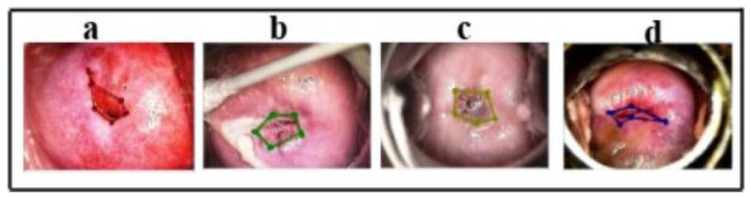
Annotation schematic diagram. Images (**a**–**d**) are the colposcopy images captured in acetic acid of different patients.

**Figure 3 cancers-17-00781-f003:**
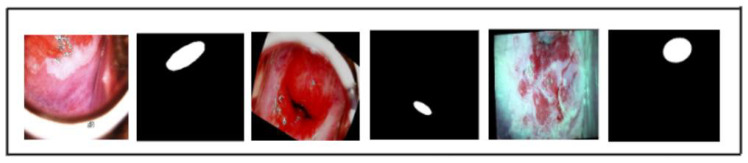
Three sample images and their corresponding masks showing the result of preprocessing.

**Figure 4 cancers-17-00781-f004:**
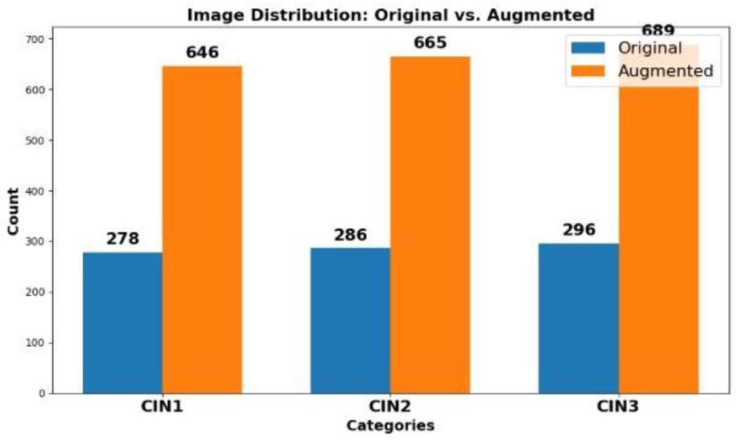
Distribution of primary dataset. The first chart represents distribution of CIN1, CIN2, and CIN3 in the original dataset. The second chart represents distribution of the three classes in the augmented dataset.

**Figure 5 cancers-17-00781-f005:**
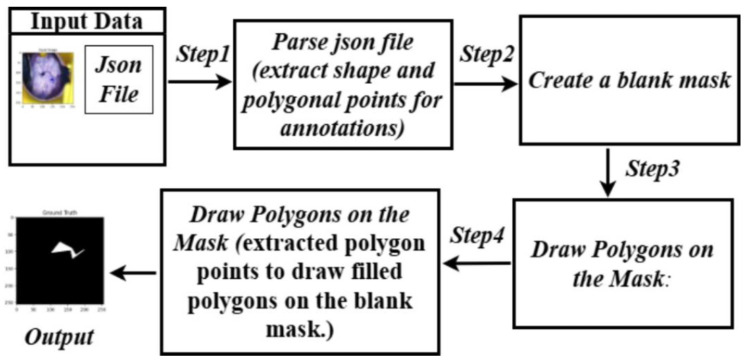
Masking architecture followed for the proposed model.

**Figure 6 cancers-17-00781-f006:**
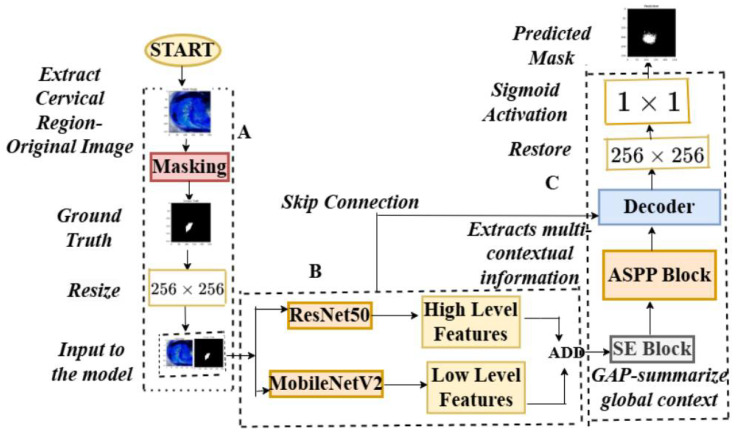
The overall architecture of the model. (**A**) The architecture of the cervical region extraction model; (**B**) feature extraction using ResNet50 and MobileNetV2; (**C**) squeeze-and-excitation block; atrous spatial pyramid pooling block, the decoder block with the predicted mask.

**Figure 7 cancers-17-00781-f007:**
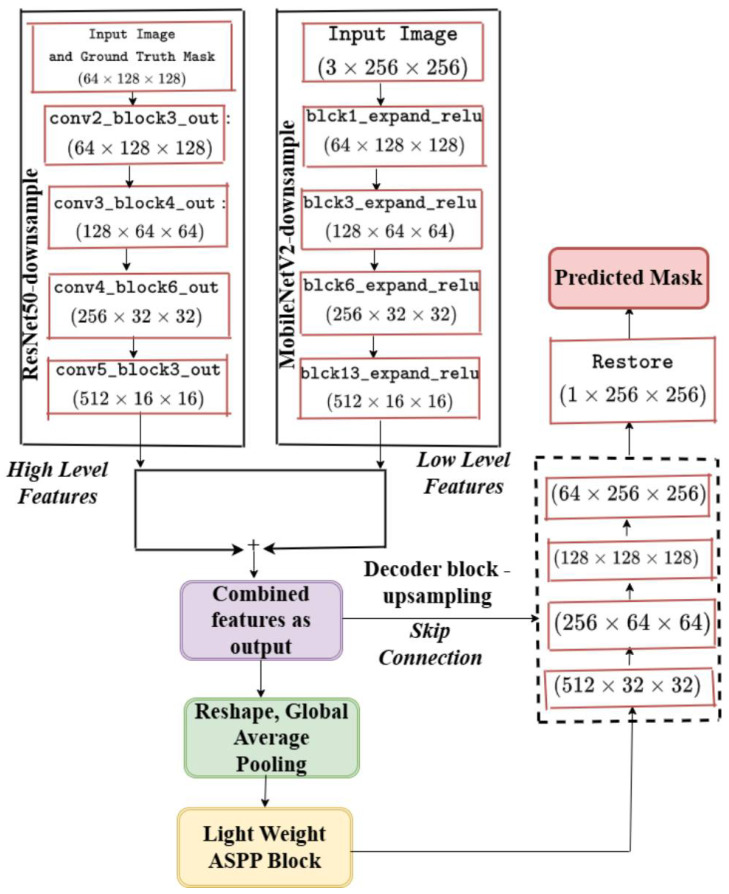
The proposed model structure in details.

**Figure 8 cancers-17-00781-f008:**
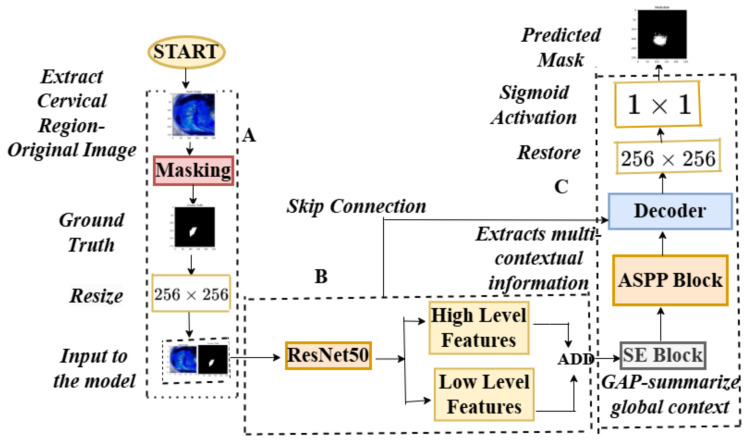
The overall architecture of the model. (**A**) The architecture of the cervical region extraction model; (**B**) feature extraction using only ResNet50; (**C**) squeeze-and-excitation block; atrous spatial pyramid pooling block, the decoder block with the predicted mask.

**Figure 9 cancers-17-00781-f009:**
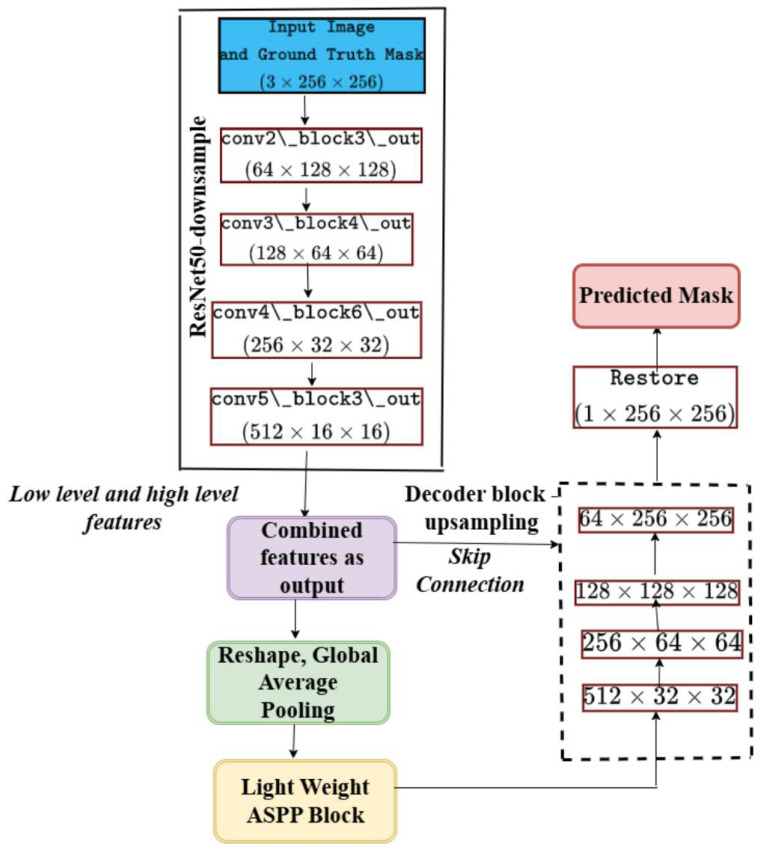
The details of the proposed model without MobileNetV2.

**Figure 10 cancers-17-00781-f010:**
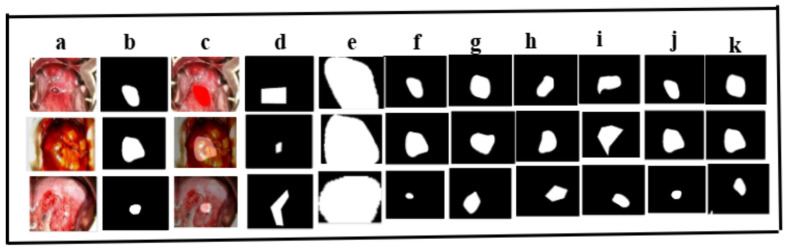
Comparison of some recent existing methods and baseline methods for segmenting colposcopic images on the test set (**a**) input image; (**b**) ground truth mask using our proposed masking algorithm; (**c**) overlay images; (**d**) W-Net [29]; (**e**) U-Net [30]; (**f**) CLS model [20]; (**g**) AUNet; (**h**) simple U-Net; (**i**) Seg Net; (**j**) proposed method; (**k**) ablation study.

**Figure 11 cancers-17-00781-f011:**
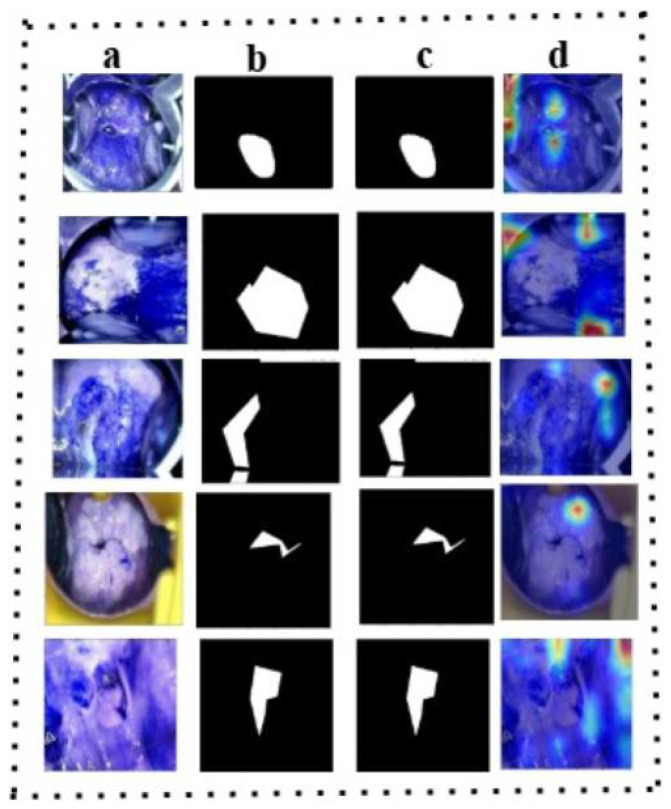
The heatmaps of the proposed model on the test samples. (**a**) Colposcopic post-acetic-acid images; (**b**) the ground truth segmentation; (**c**) the predicted segmentation; (**d**) the Gradcam++ visualization on the validation dataset.

**Figure 12 cancers-17-00781-f012:**
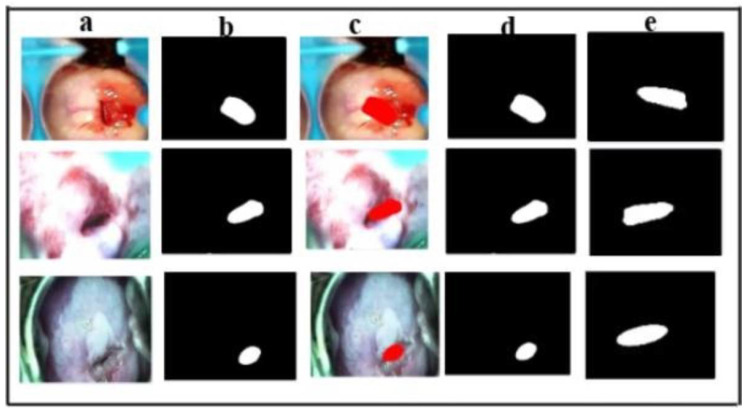
Comparison of ground truth mask and the predicted mask of the model using the dataset with and without augmentation on the test set. (**a**) input image; (**b**) ground truth mask using our proposed masking algorithm; (**c**) overlay image; (**d**) predicted mask by the model using augmented data set; (**e**) predicted mask by the model without using augmented data set.

**Figure 13 cancers-17-00781-f013:**
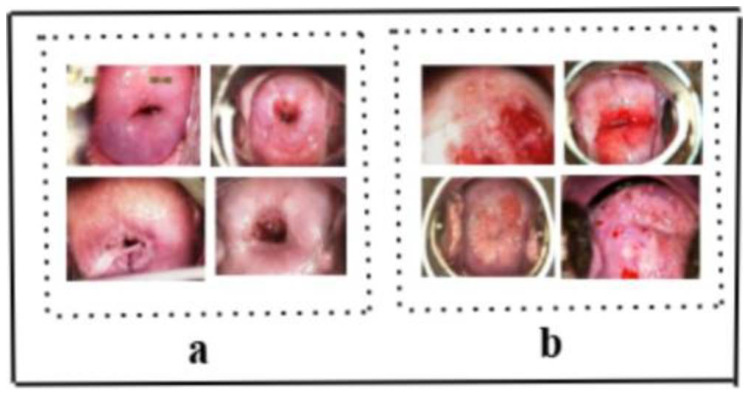
Sample images used in the loss function experiment. (**a**) Clear and simple samples. (**b**) Complicated samples with unclear border.

**Figure 14 cancers-17-00781-f014:**
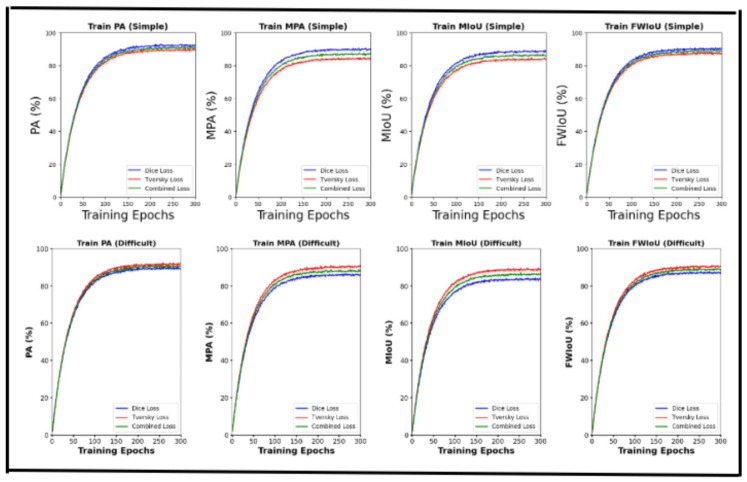
Experiment results using simple and difficult samples.

**Figure 15 cancers-17-00781-f015:**
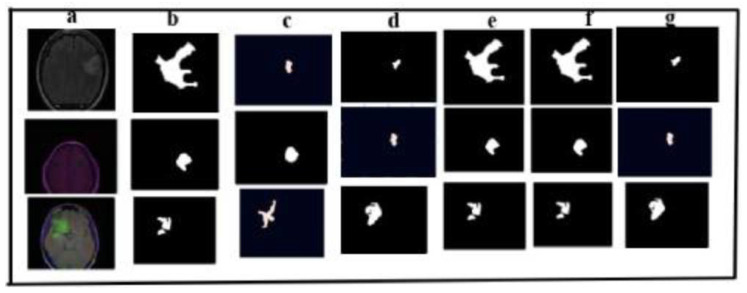
Comparison of different models for segmenting brain MRI images on the test set (**a**) input image; (**b**) manual annotation of original drawings; (**c**) [29] (**d**) [30]; (**e**) [20]; (**f**) proposed model; (**g**) ablation of the proposed model.

**Figure 16 cancers-17-00781-f016:**
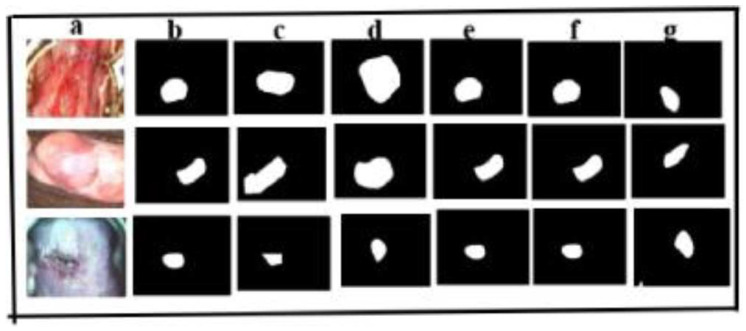
Comparison of different models for segmenting cervical cancer secondary dataset on the test samples (**a**) input image; (**b**) manual annotation of original drawings; (**c**) [29] (**d**) [30]; (**e**) [20]; (**f**) proposed model; (**g**) ablation of the proposed model.

**Table 1 cancers-17-00781-t001:** Tensor sizes when the proposed network is operating. The format in which the tensor sizes are shown is feature maps × height × width. Batch size is not taken into consideration.

Operation	Input Tensor	Output Tensor
Input Layer	3×256×256	3×256×256
ResNet50 Block 1 (conv2_block3_out)	64×128×128	64×128×128
ResNet50 Block 2 (conv3_block4_out)	128×64×64	128×64×64
ResNet50 Block 3 (conv4_block6_out)	256×32×32	256×32×32
ResNet50 Block 4 (conv5_block3_out)	512×16×16	512×16×16
MobileNetV2 Block 1 (block_1_expand_relu)	16×128×128	16×128×128
MobileNetV2 Block 2 (block_3_expand_relu)	24×64×64	24×64×64
MobileNetV2 Block 3 (block_6_expand_relu)	32×32×32	32×32×32
MobileNetV2 Block 4 (block_13_expand_relu)	96×16×16	96×16×16
Combined Feature Map	608×16×16	1024×16×16
Decoder Block 1	512×32×32	512×32×32
Decoder Block 2	256×64×64	256×64×64
Decoder Block 3	128×128×128	128×128×128
Decoder Block 4	64×256×256	64×256×256
Final Output (Segmentation Map)	64×256×256	1×256×256

**Table 2 cancers-17-00781-t002:** Experimental environment’s parameters.

Component	Name/Value
Operating system	Windows 10 Home Single Language
Python Version	3.10.12 20:22:13) [GCC 11.4.0]
TensorFlow Version	2.15.0
Keras Version	2.15.0
Hardware Accelerator	Colab Pro, v2-8 TPU
Input image size	256 × 256
Batch size	16
Epoch	300
Optimizer	Adam
Learning rate	0.0001
CPU	Intel(R) Core(TM) i5-1035G1 CPU @ 1.00 GHz, 1.19 GHz
Installed RAM	16.0 GB (15.8 GB usable)

**Table 3 cancers-17-00781-t003:** Comprehensive overview of segmentation metrics, their equations, and descriptions.

Metric	Formula	Description
Pixel Accuracy (PA)	$PA=\frac{\sum_{i=0}^{k}p_{ii}}{\sum_{i=0}^{k}\sum_{j=0}^{k}p_{ij}}$	Measures the ratio of correctly predicted pixels to the total pixel count, indicating detailed segmentation performance.
Mean-Pixel-Accuracy (MPA)	$MPA=\frac{1}{k+1}\sum_{i=0}^{k}\frac{p_{ii}}{\sum_{j=0}^{k}p_{ij}}$	Evaluates segmentation accuracy averaged across classes for enhanced pixel-level accuracy.
Mean-Intersection-over-Union (MIoU)	$MIoU=\frac{1}{k+1}\sum_{i=0}^{k}\frac{p_{ii}}{\sum_{j=0}^{k}p_{ij}+\sum_{j=0}^{k}p_{ji}-p_{ii}}$	Calculates the mean ratio of intersection to union, reflecting model overlap accuracy for segmentation tasks.
Frequency-Weighted-IoU (FWIoU)	$FWIoU=\frac{\sum_{i=0}^{k}\sum_{j=0}^{k}p_{ij}\cdot \frac{p_{ii}}{\sum_{j=0}^{k}p_{ij}+\sum_{j=0}^{k}p_{ji}-p_{ii}}}{\sum_{i=0}^{k}\sum_{j=0}^{k}p_{ij}}$	Adjusts IoU by class frequencies, improving performance for infrequent classes in segmentation.
DICE Coefficient	$DICE=\frac{2\cdot TP}{2\cdot TP+FP+FN}$	Evaluates overlap between predicted and ground truth masks, highlighting segmentation accuracy.
Hausdorff Distance (HD)	$H\left(A,B\right)=\max\left(\underset{a\in A}{\sup}\underset{b\in B}{\inf}d\left(a,b\right),\underset{b\in B}{\sup}\underset{a\in A}{\inf}d\left(a,b\right)\right)$	Measures congruity between segmentation boundaries of model predictions and ground truth, where a lower value indicates minimal discrepancy.
Cohen’s Kappa Coefficient (κ)	$\kappa =\frac{p_{o}-p_{e}}{1-p_{e}}$	Measures agreement between model-generated heatmaps and expert markings, indicating the level of agreement between AI-driven GradCam localization and human clinical interpretation. Strong agreement suggests reliable AI-aided diagnosis.

**Table 4 cancers-17-00781-t004:** Comparison of existing and baseline segmentation models on various metrics on the validation set.

Reference	Models	PA (%)	MPA (%)	MIoU (%)	FW-IoU (%)	Dice (%)	Hausdorff Distance (Pixel)	Params (Million)
[29]	W-Net	84.10	83.47	73.28	78.12	83.81	1772.75	75.50
[30]	U-Net	92.61	91.08	81.59	86.68	91.89	1629.32	44.01
[20]	CLS Model	94.77	92.87	89.34	89.94	93.32	1412.47	35.45
[31]	AUNet	90.99	89.33	84.95	86.97	89.67	1561.53	32.44
[32]	Simple-U-Net	90.12	89.69	84.75	86.91	89.98	1612.83	35.40
[33]	Seg Net	89.51	87.03	80.24	82.09	88.56	1651.91	49.07
Proposed	-	96.55	96.12	95.66	97.55	98.71	1346.75	25.32
Ablation	-	91.12	90.74	89.26	90.89	91.46	1622.52	32.19

**Table 5 cancers-17-00781-t005:** Comparison of existing and baseline segmentation models on various metrics on the validation set, including Cohen’s Kappa (κ).

Ref.	Models	PA (%)	MPA (%)	MIoU (%)	FW-IoU (%)	Dice (%)	Hausdorff Dist. (px)	κ
[29]	W-Net	84.10	83.47	73.28	78.12	83.81	1772.75	0.78
[30]	U-Net	92.61	91.08	81.59	86.68	91.89	1629.32	0.85
[20]	CLS Model	94.77	92.87	89.34	89.34	94.32	1412.47	0.88
[31]	AUNet	90.99	89.33	84.95	89.67	97.67	1561.53	0.90
[32]	Simple-U-Net	90.12	89.69	84.75	86.91	99.98	1612.83	0.91
[33]	Seg Net	89.51	87.03	80.24	82.09	88.56	1651.91	0.84
-	Proposed	96.55	96.12	95.66	97.55	98.71	1346.75	0.94
-	Proposed Ablation	91.12	90.74	89.26	90.89	91.46	1622.52	0.87

**Table 6 cancers-17-00781-t006:** Model improvement ablation experiment on the validation samples.

Models	PA/%	MPA/%	MIoU/%	FWIoU/%
ResNet50+decoder	89.61	84.73	81.59	88.68
ResNet50+MobileNetV2+decoder	90.13	86.30	84.35	89.94
ResNet50+decoder+attention module	89.86	84.25	82.42	86.96
ResNet50+MobileNetV2+decoder+ASPP module	93.02	91.68	89.58	90.16
Proposed model	96.55	96.12	95.66	97.55

**Table 7 cancers-17-00781-t007:** Comparison of the performance of the model with and without data augmentation on our primary dataset.

Data	PA/%	MPA/%	MIoU/%	FWIoU/%
Without Augmentation	94.61	93.73	92.59	94.18
With Augmentation	96.55	96.12	95.66	97.55

**Table 8 cancers-17-00781-t008:** Comparison of loss functions for colposcopic image segmentation for clear images on validation samples.

Models	PA/%	MPA/%	MIoU/%	FMIoU/%
Dice Loss	92.42	89.93	88.62	90.24
Tversky Loss	89.56	84.25	83.89	87.43
Combined Loss	90.99	87.09	86.25	88.83

**Table 9 cancers-17-00781-t009:** Comparison of loss functions for colposcopic image segmentation for complex images on the validation samples.

Models	PA/%	MPA/%	MIoU/%	FMIoU/%
Dice Loss	89.42	85.93	83.62	87.24
Tversky Loss	91.68	90.25	88.89	90.43
Combined Loss	90.55	88.09	86.25	88.83

**Table 10 cancers-17-00781-t010:** Comparison of segmentation results in brain MRI images by different existing models on the validation set.

Reference	Models	PA (%)	MPA (%)	MIoU (%)	FW-IoU (%)	Dice (%)	Hausdorff Distance (Pixel)
[29]	W-Net	87.28	84.57	82.28	82.48	86.91	1692.35
[30]	U-Net	92.89	92.11	88.39	89.98	90.89	1549.32
[20]	CLS Model	93.89	91.97	90.24	90.94	92.69	1482.37
Proposed	-	94.35	93.82	93.11	94.85	95.21	1326.55
Ablation	-	92.72	92.24	89.16	90.12	90.86	1532.82

**Table 11 cancers-17-00781-t011:** Comparison of segmentation results of cervical cancer secondary dataset with different existing models on the validation samples.

Reference	Models	PA (%)	MPA (%)	MIoU (%)	FW-IoU (%)	Dice (%)	Hausdorff Distance (Pixel)
[29]	W-Net	89.38	85.37	84.68	84.99	88.96	1642.66
[30]	U-Net	91.79	90.61	90.08	90.62	91.18	1555.52
[20]	CLS Model	94.71	92.87	91.74	91.94	93.89	1382.23
Proposed	-	95.29	94.82	93.61	94.78	94.98	1365.25
Ablation	-	92.62	91.24	90.46	90.82	91.24	1511.22

**Table 12 cancers-17-00781-t012:** Gaps identified and corrective measures.

Research Gap	Description	Corrective Measures
1. Unsupervised learning [29]	The cited architecture, the unsupervised model is designed to work with unlabeled data, making it suitable for datasets where annotations are unavailable. However, these models struggle to achieve high accuracy for segmentation tasks that require precise boundary detection, which is crucial for clinical applications.	The proposed supervised model leverages labeled data to achieve better segmentation accuracy and boundary refinement. This is particularly effective in clinical tasks like delineating cervix regions for diagnosis and treatment planning.
2. Semi-Supervised Learning [30]	The cited architecture leverages a U-Net for lesion segmentation but does not fully utilize supervised techniques. Its performance improves marginally with pre- and post-acid images but lacks advanced refinement mechanisms.	The proposed model is explicitly designed for supervised learning, leveraging the labeled data with advanced attention mechanisms and ASPP for precise feature extraction and segmentation.
3. Clinical Interpretability and Trust [20]	The CLS-Model’s reliance on faster R-CNN and its multi-step process can obscure interpretability for clinicians, making it harder to validate segmentation results.	The proposed model incorporates Grad-CAM, providing visual explanations for segmentation decisions, increasing trust and usability in clinical practice.
4. Attention U-Net in terms of clinical Workflow Integration [31]	Simple attention U-Net is relatively lightweight and can be integrated into clinical workflows. However, its simplicity might limit segmentation precision in complex cases, such as lesions with ambiguous boundaries.	The proposed model offers a balance between computational efficiency and precision. It includes interpretability (via Grad-CAM) to ensure clinician trust, while its modular design allows seamless workflow integration.
5. Simple U-Net in terms of interpretability and clinical trust [32]	Simple U-Net offers limited interpretability, which can reduce trust and usability in clinical applications. Clinicians may find it hard to validate the model’s predictions.	The proposed model includes Grad-CAM for visual explanations, enhancing interpretability and increasing clinician trust in segmentation results.
6. Seg Net in terms of boundary precision [33]	SegNet uses standard encoder–decoder upsampling with pooling indices, which may result in less precise boundary segmentation, especially for lesions with irregular shapes.	The proposed model incorporates attention modules and ASPP to refine boundaries and focus on critical features, ensuring more accurate lesion segmentation in colposcopic images.

## Data Availability

The data presented in this study are available on request from the corresponding author.

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
