# Peer review of "Attention-Enhanced Lightweight Architecture with Hybrid Loss for Colposcopic Image Segmentation"

_cancers, 2025, doi:10.3390/cancers17050781_

Round 1
Reviewer 1 Report
Comments and Suggestions for Authors
This study proposes a novel model to improve the accuracy of colposcopic imaging. In chapter 2 (Literature review) the authors should describe which databases have been searched. In addition, the conclusion section should be expanded and a possible clinical implications of these findings further described
Author Response
Comment 1: In chapter 2 (Literature review) the authors should describe which databases have been searched.
|
Response 1: Thank you for pointing this out. We agree with this comment. Therefore, we have included this (“The existing literature has been explored from databases such as IEEE Xplore, MDPI, PubMed, SpringerLink, ScienceDirect, ACM and other relevant sources.”) in the literature review section (Section 4, Page:26, first paragraph) in the manuscript and highlighted it as Reviewer 1 response in yellow.
|
Comment 2: In addition, the conclusion section should be expanded and a possible clinical implications of these findings further described.
Response 2: Agree. We did not expand the conclusion section but rather we modified it to agree with the findings of this research. We have included the actual clinical finding of this study which is tested on Pap Smear stains converted to digital images in MNJ Institute of Oncology Regional Center, Hyderabad. The result of this finding is included in the conclusion section (Section: 6, Page:32, fourth paragraph) in the manuscript and highlighted it as Reviewer 1 response in red.
|
4. Response to Comments on the Quality of English Language |
|
Point 1: The English is fine and does not require any improvement. |
|
Response 1: We thank the reviewer for this comment on English Language improvement. |
|
5. Additional clarifications |
|
[We did not include the details of the actual clinical findings but only the results in the conclusion section. If the details are required such as the experimental environment, datasets then we can provide that.] |
Reviewer 2 Report
Comments and Suggestions for Authors
please explain the email addresses u used and provide your correct ORCID ID
Author Response
|
3. Point-by-point response to Comments and Suggestions for Authors |
|
Comments 1 please explain the email addresses u used and provide your correct ORCID ID.
|
|
Response 1: Thank you for pointing this out. All the authors of this manuscript have used their personal mail ids. The database of the journal is updated with our professional emails. The ORCID ID of the corresponding author is correct. |
|
4. Response to Comments on the Quality of English Language |
|
Point 1: The English is fine and does not require any improvement. |
|
Response 1: We thank the reviewer for this comment on English Language improvement. |
|
5. Additional clarifications |
|
[We have revised few sections of this manuscript based on our understandings.] |
Reviewer 3 Report
Comments and Suggestions for Authors
The title is interesting and captivating. This research proposes advanced, efficient and accurate tools for cervical cancer diagnosis with the scope of improving diagnostic workflows and patient outcomes. The introduction provides sufficient information to introduce the reader into the subject. Still, considering the lenght of the manuscript, maybe it would be an idea to resume it. The literature review chapter should be be incorporated in discussion or added after results. Materials and methods are accurate and results provided are interesting. Discussion is elaborate enough and based on up to date references. English language used is without noticeable grammar or spelling errors, being easy to understand for a non-native English speaker. Overall, this manuscript deserves to be published
Author Response
|
Comments 1: The introduction provides sufficient information to introduce the reader into the subject. Still, considering the length of the manuscript, maybe it would be an idea to resume it.
|
|
Response 1: Thank you for pointing this out. We agree with this comment. Therefore, we have shortened the length of the introduction section, made it more precise so that it supports all the citations and the Keywords. The change is highlighted in the manuscript in Section 1, page 2.
|
|
Comments 2: The literature review chapter should be be incorporated in discussion or added after results. |
|
Response 2: Agree. The entire Literature Review Section (initially section 2) is moved after the result section (now section 4 and Gaps Identified is now section 4.1), page: 26-30.
|
|
4. Response to Comments on the Quality of English Language |
|
Point 1: The English is fine and does not require any improvement. |
|
Response 1: We thank the reviewer for this comment on English Language improvement. |
|
5. Additional clarifications |
|
[As no specific suggestion was given to us, we have placed the Literature review and gaps identified section after the Results section in response to the comment “Can be Improved” for Is the research design appropriate? ] |
Reviewer 4 Report
Comments and Suggestions for Authors
The WHO 90-70-90 strategy aims to eliminate cervical cancer in the world. Colposcopy is an important pillar to achieve this objetive.
Digital colposcopy allows to process the images and improve the diagnosis. It also offers the possibility of transmiting images to reference centers for diagnosis. This option is of capital importance in remote areas with low health resources, precisely where the greatest need exist due to prevalence of HPV.
This study presents an exhaustive review of the possibilities of colposcopy, and provides new and interesting options. This is a work aimed at technicians who are dedicated to improving medical devices, in this case the colposcope. It is difficult for physicians who are dedicated to cervical pathology and colposcopy to evaluate the methodology of the study, but I consider that it contributes great value to the study and prevention of preneoplasic lesions in the world.
Author Response
|
Comments 1 It is difficult for physicians who are dedicated to cervical pathology and colposcopy to evaluate the methodology of the study.
|
|
Response 1: Thank you for pointing this out. We acknowledge that the methodology involves advanced deep learning concepts, which may present challenges for readers primarily focused on cervical pathology and colposcopy but the methodology, results, discussion and conclusion sections are written and revised using simpler terms and linked each methodological step to its clinical relevance, emphasizing how automation enhances colposcopy efficiency and accuracy. |
|
4. Response to Comments on the Quality of English Language |
|
Point 1: The English is fine and does not require any improvement. |
|
Response 1: We thank the reviewer for this comment on English Language improvement. |
|
5. Additional clarifications |
|
[If the reviewer has specific suggestions to make the methodology more comprehensible for physicians, we would be happy to incorporate them into our manuscript.] |
Reviewer 5 Report
Comments and Suggestions for Authors
The manuscript presents a robust and innovative deep learning framework for colposcopic image segmentation that convincingly addresses several long‐standing challenges in the field. By integrating dual encoder backbones with an attention mechanism and a lightweight ASPP module, the authors have designed a system that efficiently captures both fine-grained and contextual features. The comprehensive evaluation—including rigorous ablation studies and comparisons across multiple datasets—demonstrates clear performance improvements over existing methods. The use of a hybrid loss function further enhances segmentation accuracy and model robustness, making this approach both technically sound and practically relevant.
Overall, the work is well-structured and the experimental validation is thorough, highlighting its potential for clinical application in cervical cancer screening. The inclusion of interpretability measures such as Grad-CAM++ adds an extra layer of reliability, supporting its adoption in real-world diagnostic workflows. I believe that this manuscript represents a significant contribution to medical image segmentation and should be accepted for publication.
Author Response
|
Comments 1 Overall, the work is well-structured and the experimental validation is thorough, highlighting its potential for clinical application in cervical cancer screening. The inclusion of interpretability measures such as Grad-CAM++ adds an extra layer of reliability, supporting its adoption in real-world diagnostic workflows. I believe that this manuscript represents a significant contribution to medical image segmentation and should be accepted for publication.
|
|
Response 1. We thank you for your positive feedback and for recognizing the structure, thorough experimental validation, and clinical potential of our work. We appreciate your acknowledgment of the importance of interpretability measures like Grad-CAM++, which enhance the model’s reliability and support its practical adoption in diagnostic workflows. We are grateful for your recommendation and believe that this study can contribute meaningfully to advancing AI-driven cervical cancer screening. Your insights have been invaluable in refining our work, and we look forward to its publication.
|
|
4. Response to Comments on the Quality of English Language |
|
Point 1: The English is fine and does not require any improvement. |
|
Response 1: We thank the reviewer for this comment on English Language improvement. |
|
5. Additional clarifications |